# Health-Promoting Ingredients in Goat’s Milk and Fermented Goat’s Milk Drinks

**DOI:** 10.3390/ani13050907

**Published:** 2023-03-02

**Authors:** Beata Paszczyk, Marta Czarnowska-Kujawska, Joanna Klepacka, Elżbieta Tońska

**Affiliations:** Department of Commodity and Food Analysis, The Faculty of Food Sciences, University of Warmia and Mazury in Olsztyn, 10-726 Olsztyn, Poland

**Keywords:** short-chain, branched and odd-chain fatty acids, conjugated fatty acid (CLA), folates, mineral composition, goat’s milk, goat’s milk products

## Abstract

**Simple Summary:**

Goat’s milk has beneficial effects on health condition maintenance, physiological function, and nutrition of both children and elderly people, and according to some authors, can be consumed without inducing adverse symptoms by people suffering from cow milk allergy. The content of bioactive ingredients (fatty acids, vitamins, micro, and macroelements) in goat’s milk and its products varies greatly and depends on many factors, for example, breed, animal husbandry conditions, lactation phase, and feeding. The nutritional importance of goat’s milk increases in response to the growing interest of consumers. In Poland, the commercial offer of goat’s milk products increases constantly. Local breeders and sellers appear on the market, offering their products, often of organic origin. Despite the growing popularity of goat products, the knowledge of their composition, including their health-promoting ingredients, is still incomplete. Therefore, the present study aimed to determine the contents of selected compounds, such as fatty acids, with particular emphasis on the content of *cis*9*trans*11 C18:2 (CLA) acid, minerals, and folates, in organic and commercial goat’s milk and fermented goat’s milk drinks available on the Polish market.

**Abstract:**

The present study aimed to determine the content of health-promoting compounds, and fatty acids, with particular emphasis on the content of *cis*9*trans*11 C18:2 (CLA) acid, selected minerals, folates in organic and commercial goat’s milk and fermented goat’s milk drinks. The analyzed milk and yoghurts had various contents of particular groups of fatty acids, CLA, minerals, and folates. Raw organic goat’s milk had a significantly (*p* < 0.05) higher content of CLA (3.26 mg/g fat) compared to commercial milk (2.88 mg/g fat and 2.54 mg/g fat). Among the analyzed fermented goat’s milk drinks, the highest CLA content (4.39 mg/g fat) was determined in commercial natural yoghurts, while the lowest one was in organic natural yoghurts (3.28 mg/g fat). The highest levels of calcium (1322.9–2324.4 µg/g), phosphorus (8148.1–11,309.9 µg/g), and copper (0.072–0.104 µg/g) were found in all commercial products and those of manganese (0.067–0.209 µg/g) in organic products. The contents of the other assayed elements (magnesium, sodium, potassium, iron, and zinc) did not depend on the production method, but only on the product type, i.e., the degree of goat’s milk processing. The highest folate content in the analyzed milks was found in the organic sample (3.16 µg/100 g). Organic Greek yoghurts had a several times higher content of folates, reaching 9.18 µg/100 g, compared to the other analyzed fermented products.

## 1. Introduction

Milk and dairy products contain many bioactive compounds valuable for human health and occurring in an easily absorbable form, such as fatty acids, micronutrients (especially calcium, magnesium, potassium, zinc, and phosphorus), and folates–B vitamins [1,2]. However, cow’s milk and dairy products may not be suitable for all consumers, for instance, children with protein diathesis, allergic patients, convalescents, the elderly, and people suffering from anemia, osteoporosis, and malabsorption syndrome. Goat’s milk is recommended for these groups of the population as an alternative [3]. It has a high biological value and exhibits nutritional properties. It is also characterized by a smaller diameter of fat globules than the milk of cows and contains more short-chain and polyunsaturated fatty acids [4]. Furthermore, it is digested faster than cow’s milk and hence can be recommended to people allergic to the latter [5].

Milk fat is a complex mixture of over 400 different fatty acids (FAs) with four to twenty-six carbon atoms, making it the most complex natural fat [6]. Fatty acids of milk fat are acids with different degrees of saturation (saturated, monounsaturated, and polyunsaturated). Unsaturated acids are mainly *cis* acids and *trans* isomers of these acids are found in smaller amounts. In milk fat, the fatty acids are mostly straight-chain and have an even number of carbon atoms, but fatty acids with an odd number of carbon atoms and branched-chain fatty acids are present as well. Studies conducted by many authors have indicated that the fatty acids present in milk fat may affect human health in different ways [5,7]. The main isomer of conjugated dienes in milk fat, acid *cis*9*trans*11 C18:2 (CLA), has, among others, anti-carcinogenic, anti-atherosclerotic, antioxidant, and anti-inflammatory properties [8,9,10,11,12]. *Trans*-vaccenic acid, which is the major *trans* C18:1 isomer in milk fat, shows anti-carcinogenic and anti-atherogenic effects [13]. In turn, the main MUFA of milk fat, oleic acid (*cis*9 C18:1), enhances the activity of low-density lipoprotein receptors, while decreasing the cholesterol concentration in the serum [14]. Short-chain fatty acids (SCFAs) also exhibit various bioactivities, such as anti-inflammatory and immunoregulatory effects, as well as preventive and therapeutic effects on several diseases, whereas branched-chain fatty acids (BCFAs) have been shown to elicit anti-carcinogenic effects [15,16,17]. In turn, *n*–3 PUFAs play a meaningful role in heart disease prevention and immune response improvement. Linolenic acid (C18:3) exhibits anti-carcinogenic and anti-atherogenic properties [18,19], while (*n*–6) PUFAs improve insulin sensitivity and thus, reduce the incidence of type 2 diabetes [20]. 

Mammalian milk is also a significant source of minerals in the human diet. It contains, for example, calcium, phosphorus, sodium, potassium, chlorine, iodine, magnesium, and small amounts of zinc, iron, and manganese [21,22]. In the European diet, milk is regarded as the best source of calcium, and its high bioavailability is closely correlated with a higher content of casein, lactose, and vitamin D. The absorption of calcium from milk reaches 80%, while the availability of this element from other products, for example, vegetables or cereals, is limited due to the presence of fiber, phytic and phenolic compounds, and oxalic acid [23,24,25]. Calcium deficiency in food increases the risk of developing such diseases as arterial hypertension, colorectal cancer, ischemic heart disease, osteoporosis, and kidney stones [25,26]. For consumers with malabsorption syndrome, it can be extremely beneficial to consume goat’s milk because the retention of ingredients from this product is better than from cow’s milk. In addition to calcium, this also applies to iron, which results from a greater share of nucleotides than in cow’s milk, which contributes to better absorption of this element in the intestines [27]. Iron plays an important role in supplying oxygen to human organs and muscles, and its deficiency can lead to anemia [28], which, in 2019, was diagnosed in almost 30% of women across the world aged 15–49 and in about 40% of the children aged 6–59 months (in Africa the rate was 60%) [29].

The alkaline pH of goat’s milk is associated primarily with the content of calcium, potassium, and sodium, and its salty aftertaste is due to the high content of chlorine. Compared to cow’s milk, goat’s milk has a relatively high level of magnesium, which prevents stress, increases the body’s immunity, and is a cofactor in many enzymatic reactions [21]. The contents of micro and macroelements in goat’s milk and products make it vary greatly and depend on many factors, for example, species, breed, or breeding conditions of animals, including, especially, their feeding, or lactation phase [26,27,30,31,32,33].

Folates occur naturally in foods, both those of plant and animal origin, as reduced folic acid derivatives. These essential dietary components belong to the family of water-soluble B vitamins. The name folic acid refers only to the synthetic form of the vitamin [34]. Folates are indispensable in the reactions of one-carbon groups, especially for the synthesis of DNA, RNA, and proteins. They are donors of methyl groups used in the remethylation of homocysteine to methionine [35]. Unfortunately, their deficiency is common worldwide [36]. Prolonged too low folate consumption promotes, among others, atherosclerotic processes, increasing the risk of development of cardiovascular (ischemic heart disease, stroke, thromboembolism) and neurodegenerative (Alzheimer’s, Parkinson’s) diseases. Adequate folate intake and status have also been associated with a reduced incidence of certain types of cancer (colon cancer, breast, cervix, lungs, pancreas), depressive mental disorders, and macrocytic anemia [34,35,37,38,39,40]. Young women of childbearing age are particularly vulnerable to deficiency of this vitamin. The results of epidemiological surveys indicate that an insufficient folate status in future mothers increases the probability of congenital heart defects, urinary tract abnormalities, cleft palate, limb defects, and neural tube defects (NTD) in newborns [41,42,43]. However, deficiencies can be effectively prevented by providing a folic acid dose of 400 µg per day [44]. In addition to the use of dietary supplements and foods enriched with synthetic folic acid, it is recommended, above all, to regularly consume products naturally rich in these vitamins. An example of products popular among consumers, which represent a natural source of bioactive substances, including folates, are milk and milk products [34,45]. 

In Europe, cow’s milk is most often used in dairy production. However, the popularity of goat’s milk and goat’s milk products among consumers is being observed to grow. These products are especially appreciated by pioneers of taste in regional and organic food who look for new and unique flavor sensations. The commercial offer of goat’s milk products increases year by year. Local breeders and sellers appear on the market, offering cheeses, cottage cheese, yoghurts, and kefirs made of goat’s milk, often of organic origin [46,47,48]. Despite growing interest, the knowledge of the content of many valuable compounds in goat’s milk is incomplete. Therefore, the present study aimed to determine the contents of health-promoting compounds, including fatty acids, with particular emphasis on the content of *cis*9*trans*11 C18:2 (CLA) acid, selected minerals, and folates in organic and commercial goat’s milk and fermented goat’s milk drinks. 

## 2. Materials and Methods

### 2.1. Materials

The experimental materials were organic goat’s milk, ecological fermented goat’s milk drinks, commercial goat’s milk, and commercial fermented goat’s milk drinks available on the Polish market (Table 1). The analyzed products were purchased in March 2021.

### 2.2. Methods 

#### 2.2.1. Analysis of Fatty Acids (FAs) 

The fat from the milk and yoghurts was extracted with the Folch method [49]. The IDF method was used to convert fatty acids into the corresponding fatty acid methyl esters (FAME) (ISO 15884:2002) [50]. The methyl esters obtained were then analyzed by gas chromatography (GC). Chromatographic separation was performed using a Hewlett-Packard 6890 gas chromatograph (Műnster, Germany) with a flame-ionization detector (FID) and a capillary column with a length of 100 m and an internal diameter of 0.25 mm. The liquid phase was CP Sil 88 (Chrompack, Middelburg, The Netherlands) and the film thickness was 0.20 μm., The analyses were carried out under the following temperature conditions: column temperature 60 °C; 5 °C/min increase to 180 °C; injector temperature 225 °C; and detector temperature 250 °C. The sample injection volume was 0.4 μL (split mode—50:1). Helium was used as a carrier gas at a flow rate of 1.5 mL/min. Fatty acids were identified based on a comparison of their retention times, with the retention times of fatty acid methyl esters of reference milk fat (BCR Reference Materials, CRM 164 symbol)(Aldrich, Taufkirchen, Germany), and literature data [51,52]. The *cis*9*trans*11 CLA isomer was identified using a mixture of CLA methyl esters (Sigma-Aldrich, St. Louis, MO, USA). Other positional *trans* isomers were identified using the standards of methyl esters of these isomers (Sigma-Aldrich, St. Louis, MO, USA, and Supelco, Bellefonte, PA, USA). The contents of fatty acids were expressed in mg/g fat according to the applicable standard (methyl ester of C21:0 acid, Sigma-Aldrich, St. Louis, MO, USA).

#### 2.2.2. Analysis of Minerals

The mineral content of milk and fermented dairy products was established by wet mineralization in a mixture of nitric acid and perchloric acid (3:1, *v*/*v*). The samples were mineralized in a block heating digester (DK 20, VELP Scientifica, Usmate, Italy) for 4–5 h by gradually increasing the temperature from 120 to 200 °C. The mineralized samples were transferred to 25-mL volumetric flasks, and the flasks were filled up with deionized water. Reagent samples were prepared simultaneously. The contents of copper (Cu—324.8 nm), manganese (Mn—279.5 nm), iron (Fe—248.3 nm), zinc (Zn—213.9 nm), magnesium (Mg—285.2), and calcium (Ca—422.7 nm) were determined with Flame Atomic Absorption Spectrometry (FAAS), using an iCE 3000 Series atomic absorption spectrometer (Thermo Scientific, Madison, WI, USA) equipped with a GLITE data station, deuterium lamp background correction, and various cathodic lamps. In determining the Ca content, an aqueous solution of lanthanum chloride was added to obtain an La+3 concentration of 0.5% to eliminate the effects of P. The contents of sodium (Na—589 nm) and potassium (K—766.5 nm) were determined with the Atomic Emission Spectrometry (AES) in an acetylene-air flame, using the same spectrometer, but working in emission mode. Phosphorus (P—610 nm) content was determined using the colorimetric method with ammonium molybdate (VI), sodium sulfate (IV), and hydroquinone. A VIS 6000 spectrophotometer (Thermo Scientific, Madison, WI, USA) was used for the absorbance measurements.

#### 2.2.3. Analysis of Folates

Folates: 5-methyltetrahydrofolate (5-CH_3_-H_4_folate) and tetrahydrofolate (H_4_folate) were obtained from Sigma Aldrich (St. Louis, MO, USA) and prepared according to Konings [53]. The concentration of the standards was calculated using the molar absorption coefficients given by Blakley [54]. Protease (Sigma Aldrich, St. Louis, MO, USA) was dissolved in 0.1 M phosphate buffer, pH 7.0, with 1% (*w*/*v*) sodium ascorbate and 0.1% (*v*/*v*) 2-mercaptoethanol (in the amount of 4 mg/mL). This was done just before the analysis to avoid contamination with bacteria that can synthesize folate during incubation. Some γ-Glutamyl hydrolase, from rat blood plasma, was purchased from Europa Bioproducts Ltd. (Cambridge, UK) and prepared according to Patring et al. [55]. Samples were prepared in triplicate under dim light. A 10 g sample (exact to 0.001 g) was weighed into 30-mL centrifuge flasks (30-mL PPCO Oak Ridge PPCO Nalgene centrifuge tube; Rochester, NY, USA). Then, 20 mL of an extraction buffer (0.1 M phosphate buffer, pH 7.0, with 1% (*w*/*v*) sodium ascorbate and 0.1% (*v/v*) 2-mercaptoethanol) were added. The mixture was shaken (2500 rpm/10 s Vortex 4 basic IKA Vortex 4 basic; Staufen, Germany) and heated in a water bath at 100 °C for 15 min with occasional shaking. Then, the samples were cooled in an ice bath to 20 °C, and 0.25 mL of γ-glutamyl hydrolase and 1 mL of protease were added. The samples were incubated at 37 °C for 4 h. Then, to inactivate the enzymes, the samples were heated for 5 min at 100 °C and cooled in an ice bath. Afterwards, they were centrifuged for 20 min at 12,000 rpm/4 °C (MPW-350R; Warsaw, Poland). After centrifugation, the supernatant was poured into 50-mL dark glass measuring flasks. An amount of 10 mL of the extraction buffer was added to the sediment remaining in the tubes, and the mixture was shaken and centrifuged again. The resulting supernatant was poured into the same measuring flasks. The volume of the flasks was supplemented with the extraction buffer and the whole sample was filtered through paper filters into 25-mL bottles.

Sample purification was carried out using Solid Phase Extraction (SPE) on Strong Anion Exchange (SAX) Bakerbond SPE JT cartridges (3 mL × 500 mg Solid Phase Extraction Column and PP (polypropylene), Quaternary Amine (N^+^) Anion Exchange; Philipsburg, MT, USA), as described by Jastrebova et al. [56]. Folate separation was carried out using the HPLC system (Shimadzu Nexera-i LC-2040 C plus; Shimadzu Co.; Kyoto, Japan) and the C18 LC column (150 × 4.6 mm, 3 µm, Luna 100Å; Phenomenex; Torrance, CA, USA) as described previously [57]. Identification and calculation of folate content were performed based on a standard with a known folate content (fluorescence detection, 290 nm excitation, and 360 nm emission wavelengths). 

#### 2.2.4. Statistical Analysis

The data were analyzed using one-way analysis of variance (ANOVA) to identify significant differences in the contents of fatty acids, minerals, and folates between organic and commercial goat’s milk and fermented goat’s milk drinks. Statistically different results were assessed using the Duncan’s test at a significance level of *p* < 0.05. The Statistica software package version 13.3 (StatSoft, Kraków, Poland, 2016) was used [58]. 

## 3. Results and Discussion

### 3.1. Fatty Acid Composition

Fat extracted from the analyzed goat’s milk and fermented goat’s milk drinks had various contents of fatty acids (Table 2). The major fatty acids identified in the tested samples were saturated fatty acids (SFAs). Their highest content was determined in fat from organic natural yoghurts (604.77 mg/g fat). Fat from the other analyzed yoghurts had a significantly (*p* < 0.05) lower content of SFAs. The highest content of branched-chain fatty acids (BCFAs) was found in fat from kefirs (15.54 mg/g fat), and that of odd-chain fatty acids (OCFAs) in fat from organic probiotic yoghurts (17.23 mg/g fat). The content of monounsaturated fatty acids (MUFAs) was the highest in fat from the raw milk (236.43 mg/g fat), while the lowest in fat extracted from UHT milk (160.23 and 159.34 mg/g fat, respectively). The highest content of PUFAs (29.82 mg/g fat) was determined in fat from natural yoghurts (producer 2). Significantly (*p* < 0.05) lower contents of PUFAs were determined in the other analyzed products. The present study indicates that fat extracted from commercial natural yoghurts (producer 2) had a significantly (*p* < 0.05) higher content of *n*-3 (4.69 mg/g fat) and *n*-6 PUFAs (16.80 mg/g fat) compared to the other tested products (Table 2). The lowest content of *n*-3 PUFAs was found in fat from commercial natural yoghurts (producer 1) (1.76 mg/g fat) and that of *n*-6 PUFAs in fat from organic probiotic yoghurts (10.42 mg/g fat). Determination of the contents of *n*-6 and *n*-3 fatty acids is particularly important from a nutritional point of view. An adequate supply of these acids in the diet is indicated to reduce the risk of the development of many diseases [59,60,61,62,63,64,65,66]. These acids cannot be synthesized in the body and must be derived from the diet. Although their everyday content is important, even more important is their ratio. Excessive amounts of *n*-6 PUFAs and a high *n*-6/*n*-3 ratio in the diet contribute to the pathogenesis of many diseases. An increased level of *n*-3 PUFAs and a lower *n*-6/*n*-3 ratio have a suppressive effect [65]. In our study, the *n*-6/*n*-3 ratio determined in organic goat’s milk was 5.29, and in commercial milk—5.72 (milk, producer 2) and 4.76 (milk, producer 1) (Table 2). In the tested fermented products, the *n*-6/*n*-3 ratio was the lowest in fat from organic Greek and organic probiotic yoghurts (3.00 and 3.16, respectively). The highest ratio (8.37) was determined in fat from commercial natural yoghurts (producer 1) (Table 2). According to Cossignani et al. [67], the *n*-6/*n*-3 ratio in goat’s milk was 5.8. In a previous study, Paszczyk et al. [68] showed that the ratio of these acids in goat’s milk was 6.98, and in goat yoghurts—6.04. Dairy products with a lower *n*-6/*n*-3 ratio are characterized by a better composition of fatty acids supporting the proper functioning of the body.

The acid, *cis*9*trans*11 C18:2 (CLA), which is the major CLA isomer in food, is a very important acid from a nutritional point of view. In milk and dairy products, it accounts for over 80–90% of the total CLA content [69]. The data presented in Table 2 indicate that the analyzed products had various contents of this acid. Among the goat’s milk samples, a higher CLA content was determined in fat extracted from raw organic milk (3.26 mg/g fat). Both commercial UHT kinds of milk had a significantly (*p* < 0.05) lower average CLA content, i.e., 2.88 and 2.54 mg/g fat, respectively. Cossignani et al. [67] reported a higher *cis*9*trans* 11 C18:2 acid content in goat’s milk of 3.9 mg/g fat. Differences in the quality and composition of ruminant milk are due to many factors, such as animal breed, age, lactation stadium, and health condition as well as the production system, diet, etc. [70,71,72,73,74,75,76].

Previous investigations have indicated that the composition of fatty acids, including the content of CLA, in fermented milk products may differ from the milk they were made from. The content of CLA in dairy products may be influenced by industrial technological treatments and the additives used [77,78,79,80,81]. According to research by other authors, the content of CLA in fermented dairy products is influenced by the strains of lactic acid bacteria used in the production process. Studies [82,83,84,85,86,87,88,89,90] have also shown that selected bacterial strains are capable of synthesizing CLA during fermentation, but this process can be influenced by many factors, such as the number of cells, appropriate substrate concentration, and incubation conditions. In the analyzed fermented goat’s milk drinks, the highest CLA content (4.39 mg/g fat) was determined in fat from commercial natural yoghurts (producer 2) (Table 2). In the tested organic products, Greek yoghurt produced with the vaccine containing strains of *Streptococcus thermophilus* and *Lactobacillus delbrueckii* subsp. *Bulgaricus,* and probiotic yoghurts produced with strains of *Streptococcus thermophilus*, *Lactobacillus bulgaricus*, *Bifidobacterium bifidum*, *Lactobacillus acidophilus,* and *Lactobacillus casei,* also had a high content of this acid (4.19 and 4.07 mg/g fat, respectively). The lowest CLA content, 3.28 mg/g fat, was determined in fat from organic natural yoghurt (Table 2) produced with the same strains as Greek yoghurt, which confirms the importance of the quality of the yoghurt vaccines used and the fermentation conditions applied for the synthesis of bioactive ingredients, including CLA.

The most important *trans* fatty acids (TFAs) in the human diet are monounsaturated fatty acids with 18 carbon atoms. The study showed that the average total content of *trans* C18:1 isomers in the fat extracted from the analyzed products varied (Table 2). The highest content of *trans* C18:1 was found in fat extracted from commercial natural yoghurts (17.19 and 16.61 mg/g fat), whereas the lowest was in fat from organic natural yoghurts (10.89 mg/g fat) and UHT milk (producer 2) (10.88 mg/g fat). Considering the group of *trans* C18:1, all samples contained the highest amounts of *trans*10 + *trans*11 isomers. The major TFA in milk fat is vaccenic acid (*trans*11 C18:1, VA). Its content in milk fat largely depends on the way animals are fed [69]. In ruminant milk from conventionally fed cows, this acid constitutes about 40–50% of all *trans* C18:1. Contents of other *trans* C18:1 isomers, such as *trans*9 and *trans*10, are much lower (by 5% and 10% on average, respectively) [91,92]. Milk fat also contains *trans* C18:2 isomers; however, their content is lower compared to the content of *trans* C18:1 isomers. The conducted research showed that the average content of *trans* C18:2 isomers in the fat extracted from the analyzed products was diversified, with the highest value noted for Greek yoghurt fat (4.97 mg/g fat), and significantly (*p* < 0.05) lower than for the other analyzed products (Table 2).

### 3.2. Mineral Composition

The elements that were present in the tested samples in the largest amounts were potassium, calcium, and phosphorus, with their contents noted in the range of: 1117.4–1965.3 µg/g, 1125.5–2324.4 µg/g and 1130.5–11,309.9 µg/g, respectively, regardless of product type (Table 3). The high content of these elements in goat’s milk and its products was also demonstrated by Mohammed et al. [93], who determined them in goat yoghurts at the level of 1873.3 µg/g (potassium), 1256.7 µg/g (calcium) and 923.3 µg/g (phosphorus). Bezerril et al. [94] considered these compounds to be the most important minerals found in goat’s milk products, emphasizing the role of calcium and phosphorus in the formation of the casein structure. They showed that one portion of yoghurt (100 g) would provide almost 78% of the recommended daily intake of phosphorus, 67% of calcium, and approx. 34% of potassium. These authors also claimed that goat yoghurts were a good source of sodium, which is consistent with our study results. Sodium content in all analyzed samples was in the range of 307.0–522.8 µg/g, and the next element in terms of content was magnesium (126.3–205.4 µg/g). Similar contents of these elements in goat’s milk have been reported by many authors [22,25,68,95] who point out the high bioavailability of magnesium from goat’s milk products when compared to those obtained from cow’s milk.

The minerals that were present in the tested products, in the smallest amounts, were: zinc (2.556–4.663 µg/g), copper (0.024–0.104 µg/g), manganese (0.029–0.209 µg/g), and iron (0.208–0.300 µg/g). A similar content of these elements in goat’s milk was also indicated by Zamberlin et al. [27], who determined zinc at the level of 2.420 µg/g, copper at 0.110 µg/g, manganese at 0.055 µg/g, and iron in the range from 0.360 to 0.750 µg/g, depending on milk origin and goat breed.

Analyzing the effect of milk processing on the content of the tested minerals, it should be stated that the levels of potassium, calcium, phosphorus, and magnesium were related to the degree of milk treatment. They were present in greater amounts in fermented beverages compared to milk, but this relationship was observed only in the case of commercial products (Table 3). Bergillos-Meca et al. [96] reported the similarity of quantitative changes of these components in raw and fermented goat’s milk and they showed statistically significant correlations between the levels of calcium, phosphorus, magnesium, and zinc, proving their similar susceptibility to changes during milk processing. The higher content of these elements in yoghurts obtained from goat’s milk in laboratory conditions was also demonstrated in our previous study [68]. We assayed potassium at 1797.0 µg/g in raw goat’s milk, and at 2486.2 µg/g in yoghurt obtained using a thermostatic method. These changes ranged from 1536.6 to 2178.5 µg/g in the case of calcium, from 1168.8 to 1566.2 in the case of phosphorus, and from 104.3 to 150.3 µg/g in the case of magnesium.

The levels of sodium, iron, and zinc were not related to the degree of goat’s milk processing neither in organic nor industrial production, while manganese turned out to be the element whose content was related to the degree of milk processing. This component was present in the highest amount in organic goat’s milk (0.209 µg/g), and in products obtained from it were lower by two or even three times. The content of manganese depended significantly on the f treatment method because its level was almost 10 times higher in organic milk than in commercial milk. There is little information in the available literature on the changes in manganese levels that occur during the processing of goat’s milk or milk from other mammals. Certain information on this subject was provided by Al Sidawi et al. [97], who analyzed 195 samples of cow’s milk and 25 samples of cheese from different regions of Georgia and showed that the level of this element in the analyzed milk differed 4-fold, and in cheese 3-fold, depending on the origin of the samples. Some changes in the level of manganese occurring during milk processing were also indicated by Quintana et al. [98], but the differences they identified were primarily related to the type of milk, not the degree of process advancement.

Comparing the content of mineral compounds in all organic and commercial products, regardless of their type, it should be stated that the processing method had a significant impact on the levels of calcium, phosphorus, and copper. A significantly higher level of these elements was found in all products from commercial production, and the most noticeable differences were observed in the case of phosphorus whose level was 6–10 times higher than in organic products. Such a high content of phosphorus in industrial products may be related to the level of this element in commercial feed obtained from raw materials cultivated in areas fertilized with phosphorus. This element is very important in the nutrition of mammals, because it is, in terms of content, the second highest component of the skeleton after calcium, and its deficiency causes growth inhibition, reduced productivity, and deterioration of animal fertility. Cereal grain and protein supplements, often used in industrial animal nutrition, are also good sources of phosphorus [99].

Similarly, high differences in both types of production were observed in the case of manganese, significantly higher levels of which were detected in products of organic origin, compared to commercial products (definite differences were up to 10 times). The greatest impact on the observed relationships may have been the way the goats were fed, especially the share of some cereal products or legumes in feed mixtures, which are a particularly good source of this element [100,101].

### 3.3. Folate Content

The available literature data mainly concern the content of folates in cow’s milk and cow’s milk products [88,102,103,104,105]. Table 4 presents folate contents in the tested goat’s milk and fermented goat’s milk drinks. In all tested products, the main folate form was methyl form (5-CH_3_-H_4_folate). Small amounts of H_4_folate were also determined in most of the analyzed samples. Previous studies on cow’s milk and beverages also indicated that 5-CH_3_-H_4_folate was the major or even the only identified form of folate [88,103,104,105,106,107]. Folate content in the tested material differed significantly (*p* < 0.05) among products. Among the milk samples, the highest folate content was determined in raw organic goat’s milk (3.16 µg/100 g). Both commercially sterilized milks, UHT 1 and UHT 2 had significantly (*p* < 0.05) lower folate contents, i.e., 1.87 and 2.24 µg/100 g respectively. In previous studies on cow’s milk [104,107,108], transport and storage conditions as well as high-temperature treatment, solar radiation, and acidity below neutral were indicated as possible factors causing folate losses during milk processing. Folate content in all tested milk was higher than that presented in Food Composition Tables for goat’s milk, 1 µg/100 g, but lower than that set for cow’s milk with 3.2% fat content, 5 µg/100 g [109]. Although according to other authors [104,110,111], the content of folates in cow’s milk is not high (4–10 µg/100 g) compared to other rich sources of this vitamin, such as green leafy vegetables, yeast, grains, or animal liver, due to the high frequency of consumption of dairy products, they can be a significant source of this vitamin in an everyday diet [102]. Available research studies indicate that milk and dairy products can contribute up to 15% of daily folate intake, especially in the young generation and in countries with high consumption of these products, like Sweden and the Netherlands [52,102,112].

A much higher folate content of 9.18 µg/100 g, compared to other tested fermented products, was found in organic Greek yoghurt, with a simple composition of yoghurt vaccine including *Streptococcus thermophilus* and *Lactobacillus delbrueckii* subsp. *bulgaricus* (Table 4). The lowest folate content, 1.60 µg/100 g, was determined in probiotic yoghurt. In the study of Gujska et al. [104], the folate content in fresh one-day stored yoghurts from cow’s milk did not exceed 3.5 µg/100 g, and a significant (*p* < 0.05) folate loss was observed within 34 days of cold storage. In the authors’ previous study [88] on cow’s milk yoghurts fermented with different yoghurt vaccines, higher folate levels of 4.5–10.5 µg/100 g were determined in fresh products. However, even those higher folate amounts in commercial yoghurts, when eaten in a normal daily portion, cannot meet more than 10–20% of the daily recommended intake of this vitamin. In the work on fermented milk beverages, additional attention was also paid to the possibility of folate consumption by lactic acid bacteria (LAB) during storage [113,114]. On the other hand, various studies have indicated that the selection of appropriate bacterial cultures can promote folate synthesis during fermentation [88,110,115,116,117,118]. However, this promotion depends on many factors, such as the applied starter culture (the strain, the combination of LAB), cultivation conditions, the incubation time as well as the composition of the culture medium, and the presence of folate precursors [113,115,119]. In the study of Laiño et al. [115], among different combinations, a strain of *Lactobacillus delbruecki* subsp. *bulgaricus* and *Streptococcus thermophilus* yielded the best results in milk in terms of folate content, which is consistent with the results obtained in our study for organic Greek yoghurt. Similarly, previous studies [120] on the folate content of various strains of *Saccharomyces cerevisiae* have shown that the selection of appropriate starter cultures is very important when improving the folate content in foods produced with the use of yeast. In the tested organic kefir, despite a very diverse mixture of starter cultures declared by the manufacturer, the determined folate content of 0.99 µg/100 g was the lowest among all analyzed products.

## 4. Conclusions

The research provides basic knowledge on the content of selected health-promoting components of goat’s milk and fermented goat’s milk products. Until recently, it has been believed that fat content in products is a source of energy accumulated in the fatty tissue required to build cell membranes. Today, fat’s role as a component demonstrating health-promoting properties in the human body is gaining emphasis.

The study results show that goat’s milk and fermented products are a good source of components valuable from a nutritional point of view, such as fatty acids, minerals, and folates. The amount of these nutrients depended on milk processing and production methods, which resulted in significant differences in the content of health-promoting compounds determined in goat’s milk and fermented milk drinks, as well as in organic and commercial products. The potential for obtaining increased levels of such compounds as folates and CLA was observed for organic Greek yoghurt, as well as for selected minerals, depending on the production method. However, further studies are needed to identify the impact of the quality of the raw material and the selection of starter cultures on the content of different bioactive compounds in the final product.

## Figures and Tables

**Table 1 animals-13-00907-t001:** Test material description.

Products	Characteristics of Starter Culture (According to the Information Provided by the Producer)
Organic products	
Raw milk	
Natural yoghurt	*Streptococcus thermophilus, Lactobacillus bulgaricus*
Probiotic yoghurt	*Streptococcus thermophilus*, *Lactobacillus bulgaricus*, *Bifidobacterium bifidum*, *Lactobacillus acidophilus, Lactobacillus casei*
Greek yoghurt	*Streptococcus thermophilus*, *Lactobacillus delbrueckii* subsp. *bulgaricus*
Kefir	*Streptococcus thermophilus*, *Lactococcus lactis* subsp. l*actis*, *Lactococcus lactis* subsp. c*remoris*, *Lactococcus lactis* subsp. *Lactis Biovar diacetylactis*, *Leuconostoc mesenteries* subsp. c*remoris*, *Debarymyces hansenii*, *Kluyveromyces marxianus* subsp. *marxianus*
Commercial products	
UHT milk (producer 1)	
UHT milk (producer 2)	
Natural yoghurt (producer 1)	*Lactobacillus delbruecki* sub. *bulgaricus*, *Streptococcus thermophilus*
Natural yoghurt (producer 2)	Cultures of lactic acid bacteria

**Table 2 animals-13-00907-t002:** Content of fatty acid groups (mg/g fat) in fat from goat’s milk and fermented goat’s milk drinks (Mean ± SD).

Products	ΣSFA ^1^	ΣSCFA ^2^	ΣBCFA ^3^	ΣOCFA ^4^	ΣMUFA ^5^	ΣPUFA ^6^	*n*-3	*n*-6	*n*-6/*n*-3	*trans* C18:1	*trans* C18:2	*cis*9*trans*11 C18:2(CLA)
Organic products												
Raw milk (*n* = 4)	528.40 ^d,e^ ± 17.38	105.41 ^e^ ± 2.37	12.74 ^c^ ± 0.46	14.11 ^b,c,d^ ± 0.57	236.43 ^a^ ± 7.01	21.18 ^d^ ± 0.51	2.16 ^g^ ± 0.10	11.43 ^c,d^ ± 0.35	5.29 ^c^ ± 0.10	14.66 ^b^ ± 0.39	4.32 ^b,c^ ± 0.10	3.26 ^d^ ± 0.06
Natural yoghurt (*n* = 4)	604.77 ^a^ ± 23.76	130.71 ^a^ ± 4.97	14.96 ^a,b^ ± 0.69	16.44 ^a,b^ ± 0.65	231.13 ^a,b^ ± 9.79	21.23 ^d^ ± 0.83	2.27 ^g^ ± 0.16	11.82 ^c^ ± 0.48	5.22 ^c^ ± 0.22	10.89 ^e^ ± 0.62	3.93 ^c,d^ ± 0.18	3.28 ^d^ ± 0.10
Probiotic yoghurt (*n* = 4)	584.70 ^a,b^ ± 4.58	129.23 ^a^ ± 1.77	14.54 ^a,b^ ± 0.22	17.23 ^a^ ± 0.14	180.79 ^d,e^ ± 3.60	22.08 ^d^ ± 0.34	3.30 ^c^ ± 0.04	10.42 ^e^ ± 0.19	3.16 ^g^ ± 0.03	14.79 ^b^ ± 0.26	4.29 ^b,c^ ± 0.09	4.07 ^b^ ± 0.08
Greek yoghurt (*n* = 4)	526.84 ^d,e^ ± 18.93	114.11 ^c^ ± 2.77	14.15 ^a,b,c^ ± 0.56	14.83 ^b,c^ ± 0.58	196.35 ^c^ ± 7.84	24.22 ^b,c^ ± 1.39	3.76 ^b^ ± 0.11	11.29 ^c,d^ ± 0.60	3.00 ^g^ ± 0.08	13.93 ^c^ ± 0.74	4.97 ^a^ ± 0.55	4.19 ^b^ ± 0.14
Kefir (*n* = 4)	587.52 ^a,b^ ± 27.67	122.56 ^b^ ± 4.97	15.54 ^a^ ± 0.81	16.25 ^a,b^ ± 0.98	220.82 ^b^ ± 10.24	20.96 ^d^ ± 0.85	2.66 ^e^ ± 0.15	10.86 ^d,e^ ± 0.52	4.09 ^e^ ± 0.08	12.65 ^d^ ± 0.43	3.67 ^d^ ± 0.05	3.77 ^c^ ± 0.19
Commercial products												
UHT milk (producer 1) (*n* = 4)	512.76 ^e^ ± 6.80	107.65 ^d^ ± 1.55	11.49 ^d,e^ ± 0.17	12.91 ^c,d^ ± 0.18	160.23 ^f^ ± 2.05	25.04 ^b^ ± 0.31	3.06 ^d^ ± 0.04	14.57 ^b^ ± 0.18	4.76 ^d^ ± 0.05	14.96 ^b^ ± 0.26	4.53 ^b^ ± 0.11	2.88 ^e^ ± 0.05
UHT milk (producer 2) (*n* = 4)	565.88 ^b,c^ ± 17.42	112.80 ^c,d^ ± 2.87	11.13 ^e^ ± 0.37	12.38 ^d^ ± 0.43	159.34 ^f^ ± 5.88	22.09 ^d^ ± 0.86	2.46 ^f^ ± 0.08	14.00 ^b^ ± 0.47	5.72 ^b^ ± 0.01	10.88 ^e^ ± 0.51	3.09 ^e^ ± 0.20	2.54 ^f^ ± 0.11
Natural yoghurt (producer 1) (*n* = 4)	547.98 ^c,d^ ± 9.14	127.22 ^a,b^ ± 3.20	13.83 ^b,c^ ± 2.43	12.23 ^d^ ± 2.65	190.98 ^c,d^ ± 4.10	23.80 ^c^ ± 0.42	1.76 ^h^ ± 0.06	14.68 ^b^ ± 0.28	8.37 ^a^ ± 0.29	17.19 ^a^ ± 0.25	3.54 ^d^ ± 0.30	3.82 ^c^ ± 0.08
Natural yoghurt (producer 2) (*n* = 4)	500.89 ^e^ ± 18.25	114.83 ^c^ ± 4.65	11.88 ^d,e^ ± 0.72	12.19 ^d^ ± 3.34	178.63 ^e^ ± 11.42	29.82 ^a^ ± 1.14	4.69 ^a^ ± 0.12	16.80 ^a^ ± 0.90	3.58 ^f^ ± 0.23	16.61 ^a^ ± 0.55	3.95 ^c,d^ ± 0.37	4.39 ^a^ ± 0.26

*n*—number of samples; Mean—mean value; SD—standard deviation; ^a,b,c,d,e,f,g,h^—different letters within the column indicate a statistically significant difference (*p* < 0.05), ^1^ Σ SFA—sum of all saturated fatty acids; ^2^ Σ SCFA—sum of short-chain fatty acids (C4:0–C10:0); ^3^ Σ BCFA—sum of branched-chain fatty acids; ^4^ Σ OCFA—sum of odd-chain fatty acids; ^5^ Σ MUFA—sum of monounsaturated fatty acids; ^6^ Σ PUFA—sum of polyunsaturated fatty acids.

**Table 3 animals-13-00907-t003:** Minerals content in goat’s milk and fermented goat’s milk drinks (µg/g) (Mean ± SD).

Products	Mg	Ca	Na	K	P	Cu	Mn	Fe	Zn
Organic products									
Raw milk (*n* = 4)	156.4 ^d^ ± 2.702	1125.5 ^h^ ± 5.725	347.1 ^e^ ± 6.881	1745.1 ^b,c,d^ ± 169.420	1373.7 ^e^ ± 3.731	0.040 ^e^ ± 0.004	0.209 ^a^ ± 0.005	0.286 ^c,d^ ± 0.004	4.398 ^b^ ± 0.185
Natural yoghurt (*n* = 4)	166.9 ^c^ ± 6.046	1165.4 ^g^ ± 23.985	365.3 ^d^ ± 7.781	1690.7 ^c,d^ ± 247.383	1365.1 ^e^ ± 22.682	0.040 ^e^ ± 0.002	0.118 ^b^ ± 0.003	0.267 ^e^ ± 0.011	4.239 ^c^ ± 0.095
Probiotic yoghurt (*n* = 4)	162.2 ^c^ ± 3.026	1237.6 ^e^ ± 28.320	319.6 ^f^ ± 5.278	1965.3 ^a^ ± 24.285	1143.9 ^f^ ± 39.792	0.024 ^f^ ± 0.003	0.067 ^e^ ± 0.004	0.330 ^a^ ± 0.018	3.681 ^d,e^ ± 0.067
Greek yoghurt (*n* = 4)	162.6 ^c^ ± 1.800	1195.2 ^f^ ± 14.710	365.6 ^d^ ± 7.998	1791.1 ^a,b,c^ ± 67.580	1331.1 ^e^ ± 26.655	0.052 ^d^ ± 0.004	0.091 ^d^ ± 0.004	0.275 ^d,e^ ± 0.014	3.863 ^d^ ± 0.088
Kefir (*n* = 4)	155.5 ^d^ ± 0.875	1193.9 ^f^ ± 16.575	341.8 ^e^ ± 2.113	1582.6 ^d^ ± 62.113	1130.5 ^f^ ± 18.438	0.028 ^f^ ± 0.002	0.097 ^c^ ± 0.002	0.289 ^c,d^ ± 0.011	3.678 ^e^ ± 0.065
Commercial products									
UHT milk (producer 1) (*n* = 4)	126.3 ^f^ ± 2.695	1322.9 ^d^ ± 4.343	522.8 ^a^ ± 1.749	1243.0 ^d,e^ ± 13.461	9382.9 ^c^ ± 71.219	0.104 ^a^ ± 0.001	0.029 ^g^ ± 0.0001	0.303 ^c^ ± 0.003	3.149 ^f^ ± 0.017
UHT milk (producer 2) (*n* = 4)	139.1 ^e^ ± 1.059	1554.6 ^c^ ± 9.720	307.0 ^g^ ± 1.802	1117.4 ^e^ ± 4.726	8148.1 ^d^ ± 22.030	0.072 ^c^ ± 0.002	0.030 ^g^ ± 0.0004	0.208 ^f^ ± 0.004	2.556 ^g^ ± 0.008
Natural yoghurt (producer 1) (*n* = 4)	205.4 ^a^ ± 2.869	1645.8 ^b^ ± 9.114	425.4 ^c^ ± 13.404	1931.1 ^a,b^ ± 24.932	10,591.2 ^b^ ± 232.535	0.104 ^a^ ± 0.005	0.044 ^f^ ± 0.002	0.278 ^d,e^ ± 0.001	3.729 ^d,e^ ± 0.068
Natural yoghurt (producer 2) (*n* = 4)	172.9 ^b^ ± 0.816	2324.4 ^a^ ± 21.891	454.2 ^b^ ± 11.079	1777.6 ^a,b,c^ ± 2.788	11,309.9 ^a^ ± 276.317	0.097 ^b^ ± 0.002	0.042 ^f^ ± 0.002	0.310 ^b^ ± 0.009	4.663 ^a^ ± 0.019

*n*—number of samples; Mean—mean value; SD—standard deviation; ^a,b,c,d,e,f,g^—different letters within the column indicate a statistically significant difference (*p* < 0.05).

**Table 4 animals-13-00907-t004:** Folate content in goat’s milk and fermented goat’s milk drinks (Mean ± SD).

Products	Folates (µg/100 g)
Organic products	
Raw milk	3.16 ^1,b^ ± 0.03
Natural yoghurt	2.36 ^c,d^ ± 0.09
Probiotic yoghurt	1.60 ^f^ ± 0.06
Greek yoghurt	9.18 ^a^ ± 0.42
Kefir	0.99 ^g^ ± 0.06
Commercial products	
UHT milk (producer 1)	1.87 ^e^ ± 0.01
UHT milk (producer 2)	2.24 ^d^ ± 0.05
Natural yoghurt (producer 1)	2.56 ^c^ ± 0.04
Natural yoghurt (producer 2)	3.32 ^b^ ± 0.18

^1^ Folate content is expressed as means with standard deviations from triplicates per fresh weight unit. Folates is the sum of 5-CH_3_-H_4_folate and H_4_folate expressed as folic acid content using a molar absorption coefficient given by Blakely [54]; ^a,b,c,d,e,f,g^—different letters within the column indicate a statistically significant difference at *p* < 0.05 in the Duncan test.

## Data Availability

Data is contained within the article.

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
