# Peer review of "Health-Promoting Ingredients in Goat’s Milk and Fermented Goat’s Milk Drinks"

_animals, 2023, doi:10.3390/ani13050907_

Round 1
Reviewer 1 Report
Health-Promoting Ingredients in Goat Milk and Fermented Goat Milk Drinks
The manuscript's aim of the study was to determine the content of health-promoting ingredients, fatty acids, with particular emphasis on the content of cis9trans11 C18:2 (CLA) acid, the content of selected minerals and folates in organic and commercial goat milk and fermented goat milk drinks.
The manuscript is interesting and important, both for the scientific community in the area of animal production and also for the health area, since the knowledge presented in the manuscript is often not known and disseminated.
Some information presented in the “Simple Summary” should be placed identifying the place/country of the manuscript, since this is not usually the case (Goat milk products are appreciated, among others, by amateurs of traditional and organic food, and above all by people who look for new and unique flavor sensations. The commercial offer of goat's milk products increases constantly. Local breeders and sellers appear on the market, offering their products, often of organic origin.)
In the “Simple Summary”, line 14, where it says “species”, see if it is correct or if it is better to remove it. Despite having the wild goat (Capra aegagrus), I believe that it is not suitable because the milk is not used for commercialization and, therefore, it does not fit here.
In the “Abstract”, line 28-29, put the value of CLA for commercial milk.
Some references could be used.
In general, the article is very well written, clear and with information that responds to the objectives.
Reviewer 2 Report
Statistical method were not presented precisely. It is difficult to deduce from their discription what was the factor in analysis of variance. Were organic products compared with commercial ones, or milk and fermented products within the groups of organic and commercial products. Or was the product a factor, so it would be not one-way , but multi-way analysis of variance. The lack of precision in the determination of the methods used resulted in lack of clarity in the presented results (tables 2, 3 and 4).
